Significance of NKX2-1 as a biomarker for clinical prognosis, immune infiltration, and drug therapy in lung squamous cell carcinoma

Lin Huiyue 1 linhuiyue65@163.com
Wang Juyong 1
Shi Qing 2
Wu Minmin 2
1 Oncology Department, Longhua Hospital Shanghai University of Traditional Chinese Medicine , Shanghai , China
2 Key Laboratory of Diagnosis and Treatment of Severe Hepato-Pancreatic Diseases of Zhejiang Province, The First Affiliated Hospital of Wenzhou Medical University , Wenzhou, Zhejiang , China
Jurisic Vladimir
Electronic publication date: 2024 May 1
Publication date: 2024
Volume: 12
Electronic Location ID: e17338
Received 2023 Dec 30; Accepted 2024 Apr 15
Copyright: © 2024 Lin et al.
Copyright year: 2024
Copyright holder: Lin et al.
License: This is an open access article distributed under the terms of the Creative Commons Attribution License, which permits unrestricted use, distribution, reproduction and adaptation in any medium and for any purpose provided that it is properly attributed. For attribution, the original author(s), title, publication source (PeerJ) and either DOI or URL of the article must be cited.
License URL: https://creativecommons.org/licenses/by/4.0/

Keywords: NKX2-1/TTF-1, Lung squamous cell carcinoma, Prognosis, Immune infiltration, Therapy

Funding: The authors received no funding for this work.

==============================
Background

This study was performed to determine the biological processes in which NKX2-1 is involved and thus its role in the development of lung squamous cell carcinoma (LUSC) toward improving the prognosis and treatment of LUSC.

Methods

Raw RNA sequencing (RNA-seq) data of LUSC from The Cancer Genome Atlas (TCGA) were used in bioinformatics analysis to characterize NKX2-1 expression levels in tumor and normal tissues. Survival analysis of Kaplan–Meier curve, the time-dependent receiver operating characteristic (ROC) curve, and a nomogram were used to analyze the prognosis value of NKX2-1 for LUSC in terms of overall survival (OS) and progression-free survival (PFS). Then, differentially expressed genes (DEGs) were identified, and Kyoto Encyclopedia of Genes and Genomes (KEGG), Gene Ontology (GO), and Gene Set Enrichment Analysis (GSEA) were used to clarify the biological mechanisms potentially involved in the development of LUSC. Moreover, the correlation between the NKX2-1 expression level and tumor mutation burden (TMB), tumor microenvironment (TME), and immune cell infiltration revealed that NKX2-1 participates in the development of LUSC. Finally, we studied the effects of NKX2-1 on drug therapy. To validate the protein and gene expression levels of NKX2-1 in LUSC, we employed immunohistochemistry(IHC) datasets, The Gene Expression Omnibus (GEO) database, and qRT-PCR analysis.

Results

NKX2-1 expression levels were significantly lower in LUSC than in normal lung tissue. It significantly differed in gender, stage and N classification. The survival analysis revealed that high expression of NKX2-1 had shorter OS and PFS in LUSC. The multivariate Cox regression hazard model showed the NKX2-1 expression as an independent prognostic factor. Then, the nomogram predicted LUSC prognosis. There are 51 upregulated DEGs and 49 downregulated DEGs in the NKX2-1 high-level groups. GO, KEGG and GSEA analysis revealed that DEGs were enriched in cell cycle and DNA replication.The TME results show that NKX2-1 expression was positively associated with mast cells resting, neutrophils, monocytes, T cells CD4 memory resting, and M2 macrophages but negatively associated with M1 macrophages. The TMB correlated negatively with NKX2-1 expression. The pharmacotherapy had great sensitivity in the NKX2-1 low-level group, the immunotherapy is no significant difference in the NKX2-1 low-level and high-level groups. The analysis of GEO data demonstrated concurrence with TCGA results. IHC revealed NKX2-1 protein expression in tumor tissues of both LUAD and LUSC. Meanwhile qRT-PCR analysis indicated a significantly lower NKX2-1 expression level in LUSC compared to LUAD. These qRT-PCR findings were consistent with co-expression analysis of NKX2-1.

Conclusion

We conclude that NKX2-1 is a potential biomarker for prognosis and treatment LUSC. A new insights of NKX2-1 in LUSC is still needed further research.

Introduction

Lung cancer is the world’s most common and deadliest malignant respiratory tumor, with 2.2 million estimated cases based on the 2020 report of the International Agency for Research on Cancer (https://gco.iarc.fr/, accessed on 15 September 2022). Small cell lung cancer (SCLC, 15%) and non–small cell lung cancer (NSCLC, 85%) are the two main types of lung cancer, with NSCLC patients demonstrating lower rates of overall survival and 5-year survival (Gao et al., 2019; Yu et al., 2021). NSCLC is divided into LUAD and LUSC according to pathogenesis and histological morphology (Gao et al., 2019). LUSC, which comprises 30% of cases of NSCLC, has a high rate of metastasis and recurrence (Qian et al., 2016). The current clinical first-line therapy for LUSC involves use of immune checkpoint inhibitors in combination with carboplatin and paclitaxel (Fan, Yang & Chang, 2019). Although several therapies are confirmed to be beneficial for LUSC in prolonging progression-free survival, the clinical benefits for LUSC patients remains limited (Pan et al., 2021; Gao et al., 2020). Therefore, an investigation into novel biomarkers is required to improve diagnoses and treatment of LUSC patients. Many studies on LUSC-related genes and prognostic markers have reported that the molecular mechanisms underlying the pathogenesis and progression of LUSC are not clear (Gao et al., 2020). Thus, clarification of these mechanisms is required for the development of new promising biomarkers or potential drug treatments, which are urgently needed.

NK2 homeobox 1 (NKX2-1), also known as thyroid transcription factor-1 (TTF-1), is a member of the NKX2 family of homeodomain-containing transcription factors (Yang et al., 2012). NKX2-1 regulates normal lung development and morphogenesis, especially in lung epithelial cell differentiation, and was demonstrated to be important for the occurrence of lung cancer (Yang et al., 2012; Guan et al., 2021). An independent study showed that NKX2-1 possibly regulated the adeno-to-squamous transdifferentiation to shape the tumor microenvironment or affected immune cell types shaping the correspongding tumor microenvironment, then determined tumor phenotype (Mollaoglu et al., 2018). NKX2-1/TTF-1 has been used as a diagnostic marker for LUAD and SCLC and is approximately 70% positive for LUAD, in which it is an indicator of favorable prognosis (Yang et al., 2012; Park et al., 2019). Although the expression of NKX2-1/TTF-1 in LUSC appears to very low or indetectable, a couple of studies have demonstrated a close connection exists between NKX2-1 and LUSC (Yu et al., 2021; Yang et al., 2012). As there are currently few studies on this association, we sought to examine this further by downloading RNAseq data on LUSC from TCGA, which contains the genetic profiles of more than 20 different types of tumors (Chandran et al., 2016). In the present study, the database was analyzed to characterize the expression levels of NKX2-1 in LUSC compared with normal tissues. We then explored the relationship between NKX2-1 expression and clinical characteristics, TMB, the infiltration of immune cells, immune checkpoint genes, the TME, and the pharmacotherapy response. Furthermore, we studied the co-expression of DEGs in NKX2-1 high-level and low-level groups in conjunction with GO and KEGG analyses of DEGs to identify significant biological functions and pathways. In summary, the results of this study may provide new clues to understand the underlying molecular mechanisms of NKX2-1 in LUSC and its influence on immune landscapes, TME, and the pharmacotherapy of LUSC.

Materials and Methods

TIMER database

TIMER (http://timer.cistrome.org, accessed on 29 July 2022) is a comprehensive resource for the systematic analysis of immune infiltrates, which includes more than 10,000 samples across 32 cancer types from TCGA (Li et al., 2016; He et al., 2021). We used TIMER to explore the mRNA transcriptional level of NXK2-1 in various cancer types.

Data processing

RNA-seq data profiles and relevant clinical information were downloaded from the TCGA database (https://portal.gdc.cancer.gov, accessed on 29 July 2022), which included 502 LUSC and 49 normal lung tissue samples. In our study, RNA-seq data were processed and normalized using the ‘limma’ package in R. Expression levels were quantified as fragments per kilobase of transcript per million mapped reads (FPKM), which were then transformed to log2 fold-change (Log2FC) values for subsequent analysis (Song, Li & Zhang, 2017). The missing OS values of patients were excluded to reduce statistical bias. All downloaded files were calibrated, normalized, and log2-transformed by R software (version 4.1.3; R Core Team, 2022).

Identification and validation of NKX2-1 gene expression in TCGA database

The identification of DEGs was performed between normal tissue and tumor tissue using the “limma” R package, with absolute Log2FC > 1 and false discovery rate (FDR) <0.05. A heatmap plot was drawn to exhibit the expression difference of other genes between NKX2-1 high-level and low-level groups via the pheatmap R package.

Functional and pathway enrichment analysis

The R package “clusterProfiler” was used to performed function and pathway analyses in both NKX2-1 groups according to GO and KEGG. GSEA was performed in R software with c2.cp.kegg.v7.0 symbols.gmt as the reference gene set and the top five enrichment analysis results were visualized, with p < 0.05 indicating the significant enrichment of functional annotations.

The relationship between NKX2-1 and other genes

The “limma”, “ggplot2”, “ggpubr”, “ggplot2”and “ggExtra” packages in R were used to analyze the relationship between co-expressed genes and NKX2-1 using the Pearson method. Circos was used to intuitively exhibit the correlation between co-expression genes, with red representing positive and green representing negative correlation.

Predictive nomogram design

A nomogram was constructed and predicted based on the age, gender, stage, and risk score using the “rms” package and Cox regression model to predict the OS of LUSC patients at 1, 3, and 5 years. A calibration plot was used to evaluate the nomogram, which was based on Harrell’s concordance index (C-index). “Points” was the scoring scale for each factor, and “total points” was the scale for total score. Based on the total score of the patient, the 1-, 3- and 5-year survival rate was inferred.

Correlation analysis of NKX2-1 expression in TME and TMB

The TME, contains tumor cells, surrounding immune, and stroma cells (Arneth, 2019). The R package “ESTIMATE” was used to compute the StromalScore, ImmuneScore, and ESTIMATEScore (Yoshihara et al., 2013). CIBERSORT was applied to estimate the proportion of 22 immune cells for each sample in both NKX2-1 groups (Friedman, Hastie & Tibshirani, 2010). The p-values were based on the Wilcoxon signed-rank test, and p < 0.05 was considered to indicate a statistically significant difference. The correlation of NKX2-1 expression with immune cells was conducted by using Pearson correlation analysis in the R package. The Pearson correlation test was used to investigate the correlation between NKX2-1 and 17 immune checkpoint-related genes (such as BTLA, TNFSF14, CD80, and CD244), with results visualized using the pheatmap R package. The tumor mutational burden (TMB) is defined as the total number of base mutations per million cells in the tumor, and represents the number of mutations per megabase (Mut/Mb) of DNA in cancer, that is assessed by whole exome sequencing (WES), the systematic sequencing of all exons (Addeo et al., 2021). The correlation of NKX2-1 expression with TMB was analyzed by the Spearman correlation test in R software.

Therapy in NKX2-1 high-level and low-level groups

Pharmacotherapy sensitivity analysis was based on the half-maximal inhibitory concentration (IC50), an indicator of the rate of response of tumor cells to pharmacotherapy. The “pRRophetic” package was used to predict the drug sensitivity of the two NKX2-1 groups (Geeleher, Cox & Huang, 2014a, 2014b). Immunotherapy data were obtained from the TCIA website (https://tcia.at/) and visualized in a violin plot through R software.

Verification of NKX2-1 in lung cancer

The GEO database (https://www.ncbi.nlm.nih.gov/geo/) served as the validation set. GEO datasets (GSE67061, GSE84784, GSE101420) were calibrated and normalized using R software. Mining analysis of the NKX2-1 gene was performed. Verification of NKX2-1 protein expression in LUSC and LUAD was conducted using The Human Protein Atlas (HPA, https://www.proteinatlas.org/). Additionally, human LUAD cell lines (PC-9) and human LUSC cell lines (H520) were procured from the Shanghai Institute of Biosciences and Cell Resources Center (Chinese Academy of Sciences, Shanghai, China). PC-9 cells were cultured in Dulbecco’s Modified Eagle Medium (DMEM, Thermo Fisher Scientific, Waltham, MA, U.S.), and H520 cells were cultured in Roswell Park Memorial Institute (RPMI)-1640 medium (Thermo Fisher Scientific, Waltham, MA, USA), both supplemented with 10% fetal bovine serum (FBS; Thermo Fisher Scientific). Cells were maintained in a humidified cell incubator at 37 °C with 5% CO2. Total RNA was isolated from the cells using TRIzol Reagent (Invitrogen, Waltham, MA, USA), and RNA concentration was determined using a DS-11 Spectrophotometer (DeNovix, USA, Wilmington, DE, USA). Reverse transcription was performed using HiScript III RT SuperMix for qPCR (Vazyme, China), followed by qRT-PCR using Taq Pro Universal SYBR qPCR Master Mix (Vazyme, China) on the CFX96 Real-Time System (Bio-Rad, California, U.S.). Primers were obtained from Sangon Biotech (Shanghai, China), with their sequences shown in Table 1. GAPDH was used as the reference gene for normalization. The expression differences of genes were calculated using the 2-Δct method (Zhang et al., 2017).

Table 1 The primer sequences of genes.

Gene	Forward primer	Reverse primer	
NKX2-1	AGCACACGACTCCGTTCTC	GCCCACTTTCTTGTAGCTTTCC	
NAPSA	TCTTCGTACCTCTCTCGAACTAC	GGCAACAGTGAAGTTTTGTGG	
SFTPD	CCTTACAGGGACAAGTACAGCA	CTGTGCCTCCGTAAATGGTTT	
TRIM29	CTGTTCGCGGGCAATGAGT	TGCCTTCCATAGAGTCCATGC	
IRF6	CCCCAGGCACCTATACAGC	TCCTTCCCACGGTACTGAAAC	
GAPDH	GGAGCGAGATCCCTCCAAAAT	GGCTGTTGTCATACTTCTCATGG	

Statistical analysis

The statistical analyses were performed in R (version 4.1.3; R Core Team, 2022), which included the Wilcoxon, Kruskal–Wallis, and chi-square statistical tests. The relationship between NKX2-1 expression and LUSC clinicopathological features was shown as box plots using the “limma” and “ggpubr” R packages. The data from the TCGA database were divided into NKX2-1 high-level and low-level groups based on the median expression level. Kaplan–Meier survival analysis was performed using the R packages “survminer” and “survival” to assess the differences in OS and PFS between the groups. Univariate and multivariate analysis were performed using the Cox proportional hazards regression model to identify significant factors. The time-dependent ROC curve analysis and area under the curve (AUC) were plotted by using the “timeROC” package in R to evaluate the predictive accuracy of the NKX2-1 expression at different endpoints (1, 3, or 5 years) of the prognostic risk score mode. Statistical analyses were conducted using SPSS 23.0 statistics software (SPSS, USA). Student’s t-test was utilized to determine differences between two experimental groups, while one-way ANOVA was employed for multiple group comparisons. A p-value < 0.05 was considered statistically significant.

Results

NKX2-1 mRNA expression levels in various cancers

NKX2-1 expression levels in various cancers were explored using TIMER. The results reveal that NKX2-1 expression levels were significantly lower in LUSC but significantly higher in thyroid carcinoma (THCA). Although NKX2-1 expression levels were very low in these cancers, its expression levels were significantly different in bladder urothelial carcinoma (BLCA), colon adenocarcinoma (COAD), glioblastoma multiforme (GBM), head and neck squamous cell carcinoma (HNSC), HPV-positive HNSC and HPV-negative HNSC, kidney renal papillary cell carcinoma (KIRP), prostate adenocarcinoma (PRAD), skin cutaneous melanoma (SKCM), SKCM-metastasis, and uterine corpus endometrial carcinoma (UCEC) (Fig. 1A). We analyzed the NKX2-1 expression data from TCGA to further characterize NKX2-1 expression in LUSC. According to the paired and unpaired results, its expression in LUSC tumor tissue was significantly lower than that in normal tissue (Figs. 1B, 1C).

Figure 1 NKX2-1 mRNA expression levels in various cancers.

NKX2-1 expression levels in (A) various cancer types, (B) LUSC vs. normal tissue, and (C) in LUSC vs. normal tissue with NKX2-1 paired expression analysis. Characterization is based on the tumor immune estimation resource (TIMER) database. *p < 0.05, ***p < 0.001.

Evaluation of clinical parameters and development of a prognostic prediction model for NKX2-1 in LUSC patients

The correlations between NKX2-1 gene expression and clinical characteristics, including age, gender, stage, and TNM classifications, were explored. The results showed that NKX2-1 expression did not significantly differ according to age, T classification, or M classification (p > 0.05). NKX2-1 expression significantly differed according to gender and for Stage I vs. Stage II, Stage II vs. Stage III, and N0 vs. N1 (p < 0.05) (Figs. 2A–2F). In addition to stage (p < 0.05), there were no significant differences in the NKX2-1 high-level and low-level groups due to age, gender, TNM classification, race, smoking status, site of tumor and treatment (p > 0.05) (Table 2). We found that LUSC patients with higher NKX2-1 expression had shorter OS (p = 0.015) and PFS (p = 0.036) (Figs. 2G, 2H). Further, we performed univariate and multivariate Cox regression analyses. NKX2-1 was significantly associated with OS in univariate (HR = 1.462, 95% CI [1.082–1.976], p = 0.013) and multivariate (HR = 1.495, 95% CI [1.104–2.025], p = 0.009) Cox regression analysis (Table 3). This suggests that NKX2-1 is an independent prognostic factor. ROC curves were constructed to evaluate the prognostic accuracy, and the 1-,3-, and 5-year AUC values of NKX2-1 were 0.574, 0.564, and 0.542, respectively (Fig. 2I). We constructed a nomogram to predict LUSC prognosis precisely (Fig. 2J). The sum of four points could be obtained according to the NKX2-1 expression level, gender, age, and stage, with each total point corresponding to the predicted 1-, 3-, and 5-year OS. Good agreement was observed between the observed and predicted OS rates at 1, 3, and 5 years in plots (Fig. 2K). These results demonstrate that NKX2-1 expression has a certain reference value for LUSC prognosis.

Figure 2 Evaluation of clinical parameters and development of a prognostic prediction model for NKX2-1 in LUSC patients.

Association between NXK2-1 expression and (A) age, (B) gender, (C) stage, (D) tumor, (E) metastasis, and (F) node in LUSC patients. (G) Kaplan–Meier curves for OS and (H) PFS according to NKX2-1 mRNA expression levels, stratified into high and low levels based on the median (p < 0.05). (I) ROC curves for 1-, 3-, and 5-year OS. (J) nomogram predicting the probability of OS at 1, 3, and 5 years. (K) Calibration plot predicting the agreement between observed and predicted rates of OS at 1, 3, and 5 years. *p < 0.05, **p < 0.01.

Table 2 Relationship of clinical parameters for LUSC patients in NKX2-1 high-level and low-level groups.

Characteristics	NKX2-1 low-level	NKX2-1 high-level	p-value	
Number	251	250		
Age, N (%)			0.329	
<65	93 (37.1)	77 (30.8)		
≥65	154 (61.4)	168 (67.2)		
NA	4 (1.6)	5 (2.0)		
Gender, N (%)			0.097	
Male	194 (77.3)	177(70.8)		
Female	57 (22.7)	73(29.2)		
Race, N (%)			0.569	
White	181 (72.1)	168 (67.2)		
Asia	4 (1.6)	5 (2.0)		
Black of african american	12 (4.8)	18 (7.2)		
NA	54 (21.5)	59 (23.6)		
Smoking status, N (%)			0.112	
Smoker	206 (82.1)	218 (87.2)		
Non-smoker	45 (17.9)	32 (12.8)		
Site of tumor, N (%)			0.193	
Upper lobe	122 (48.6)	138 (55.2)		
Middle lobe	5 (2.0)	11 (4.4)		
Lower lobe	92 (36.7)	81 (32.4)		
Main bronchus	4 (1.6)	3 (1.2)		
Overlopping lesion of lung	6 (2.4)	2 (0.8)		
Lung NOS	22 (8.8)	15 (6.0)		
Stage, N (%)			0.070	
I	109 (43.4)	135 (54.0)		
II	96 (38.8)	66 (26.4)		
III	40 (15.9)	44 (17.6)		
IV	4 (1.6)	3 (1.2)		
NA	2 (0.8)	2 (0.8)		
T classification, N (%)			0.649	
T1	53 (21.1)	61 (24.4)		
T2	146 (58.2)	147 (58.8)		
T3	39 (15.5)	32 (12.8)		
T4	13 (5.2)	10 (4.0)		
M classification, N (%)			0.984	
M0	205 (81.7)	206 (82.4)		
M1	4 (1.6)	3 (1.2)		
MX	40 (15.9)	39 (15.6)		
NA	2 (0.8)	2 (0.8)		
N classification, N (%)			0.116	
N0	151 (60.2)	168 (67.2)		
N1	77 (30.7)	54 (21.6)		
N2	20 (8.0)	20 (8.0)		
N3	1 (0.4)	4 (1.6)		
NX	2 (0.8)	4 (1.6)		
Treatment, N (%)			0.073	
No treatment	125 (49.8)	144 (57.6)		
Pharmaceutical therapy	59 (23.5)	38 (15.2)		
Radiation therapy	9 (3.6)	14 (5.6)		
Pharmaceutical and radiation therapy	28 (11.2)	20 (8.0)		
NA	30 (12.0)	34 (13.6)		

Table 3 Univariate and multivariate Cox regression hazard analyses of NKX2-1 expression.

Chracteristics	Univariate analysis	Multivariate analysis	
HR (95% CI)	p value	HR (95% CI)	p value	
Age (<65 vs. ≥65)	1.440 [1.030–2.014]	0.033	1.484 [1.058–2.081]	0.22	
Gender (Male vs. Female)	0.736 [0.511–1.059]	0.099			
Smoking status (No-Smoker vs. Smoker)	0.935 [0.586–1.491]	0.778			
Site of tumor (Upper lobe vs. Other sites)	0.932 [0.846–1.027]	0.155			
Stage (Stage I vs. Stage II-IV)	1.230 [1.026–1.474]	0.025	1.295 [1.082–1.551]	0.005	
T classification (T1 vs. T2-4)	1.218 [0.998–1.487]	0.053			
M classification (M0 vs. M1)	2.431 [0.897–6.586]	0.081			
N classification (N0 vs. N1-3)	1.118 [0.901–1.387]	0.311			
NKX2-1 (Low vs. High)	1.462 [1.082–1.976]	0.013	1.495 [1.104–2.025]	0.009	

Comparison analysis in NKX2-1 high-level and low-level groups and co-expression analysis of NKX2-1

We used the R software (version 4.1.3; R Core Team, 2022) to perform a comparative study between NKX2-1 high-level and low-level groups. A total of 51 upregulated DEGs and 49 downregulated DEGs in the NKX2-1 high-level groups were plotted in a heatmap (Fig. 3A). NKX2-1-AS1, SLC22A31, NAPSA, SFTA2, C16orf89, SFTPD correlated positively and TRIM29, LINC01980, GJB5, KRT5, and IRF6 correlated negatively with NKX2-1 in terms of expression (Figs. 3B–3L, p < 0.05). The top 11 co-expressed genes are shown in the Circos plot, light pink represents positive correlation, light blue represents negative correlation (Fig. 3M).

Figure 3 Comparison analysis in NKX2-1 high-level and low-level groups and co-expression analysis of NKX2-1.

(A) Heatmap showing the top 100 genes with the highest expression variation of DEGs in the NKX2-1 high-level and low-level groups; graded color scale of blue to red represents levels of gene expression. NKX2-1 expression correlated positively with (B) NKX2-1-AS1, (C) SLC22A31, (D) NAPSA, (E) SFTA2, (F) C16orf89, and (G) SFTPD and correlated negatively with (H) TRIM29, (I) LINC01980, (J) GJB5, (K) KRT5, and (L) IRF6 expression. (M) The top 11 significant genes that were either positively or negatively correlated with NKX2-1 shown in a Circos plot.

Functional enrichment analyses of DEGs in NKX2-1 high-level and low-level groups

GO function and KEGG pathway enrichment analysis were used to reveal the function and mechanisms of 729 DEGs in NKX2-1 high-level and low-level groups. The GO terms were divided into biological process (BP), cellular component (CC) and molecular function (MF) ontologies. The GO analysis results indicate that the DEGs are mainly enriched in the following BP categories: humoral immune response, sodium ion transport, negative regulation of peptidase activity, antimicrobial humoral response, antibacterial humoral response, and bicarbonate transport. The analysis shows that the DEGs were significantly enriched in the CC categories of collagen-containing extracellular matrix, apical part of cell, apical plasma membrane, blood microparticle, multivesicular body, and lamellar body. DEGs enriched in MF were mainly enriched in the categories of metal ion transmembrane transporter activity, passive transmembrane transporter activity, ion channel activity, cation channel activity, gated channel activity, and potassium channel activity (Figs. 4A–4C). In addition, the results of the KEGG pathway analysis indicated that 729 DEGs were enriched in neuroactive ligand–receptor interaction, complement and coagulation cascades, cytokine–cytokine receptor interaction, bile secretion, and cAMP signaling pathway (Figs. 4D, 4E).

Figure 4 Functional enrichment analyses of DEGs in NKX2-1 high-level and low-level groups.

(A) Circle plot of enriched biological process. The outer ring represents GO terms, with different colors distinguishing categories of biological process (BP), cellular component (CC), and molecular function (MF). The second ring within the outer ring shows the number of enriched genes. The third ring represents the number of enriched DEGs. The fourth ring represent the gene ratio. (B–E) Bar and bubble plots showing KEGG and GO enrichment analysis, respectively. Circle sizes represent the number of genes in each functional class. The graded color scale of blue to red represents the alterations of p values.

GSEA identifies DEG-related signaling pathways in NKX2-1 high-level and low-level groups

To explore the mechanisms in which DEGs are involved in LUSC, we identified pathways that showed significant differences between the NKX2-1 high- and low-expression groups by conducting GSEA (Fig. 5). KEGG_CELL_CYCLE, KEGG_PPAR_SIGNALING_PATHWAY, KEGG_DNA_REPLICATION, and KEGG_HOMOLOGOUS_RECOMBINATION were active in the low-level group, whereas KEGG_OLFACTORY_TRANSDUCTION was active in the high-level group.

Figure 5 GSEA identifies DEG-related signaling pathways in NKX2-1 high-level and low-level groups.

GSEA enrichment analysis of DEGs in NKX2-1 high-level and low-level groups.

Immune infiltration analysis and tumor mutational burden of NKX2-1 expression

We explored the correlation of NKX2-1 expression level in immunity, ESTIMATE, and CIBERSORT. The ESTIMATE results indicate that the stromal, immune, and estimate scores were lower in the NKX2-1 low-level group than the NKX2-1 high-level group (Fig. 6A). Moreover, we found that infiltration levels for mast cells resting, neutrophils, monocytes, T cells CD4 memory resting, and macrophages M2 were significantly higher in the NKX2-1 high-level group than in the NKX2-1 low-level group, and the macrophage M1 infiltration level was comparatively higher in the NKX2-1 low-level group (Fig. 6B). Further, we performed correlation analysis between NKX2-1 expression and immune infiltration cells. The results show that NKX2-1 expression was positively associated with mast cells resting, neutrophils, monocytes, T cells CD4 memory resting, and M2 macrophages but negatively associated with M1 macrophages (Figs. 6C–6I). Interestingly, we found that the tumor mutational burden correlated negatively with NKX2-1 expression (Fig. 6J). We also analyzed the relationship between NKX2-1 and immune checkpoint genes, which correlated positively (Fig. 6K).

Figure 6 Immune infiltration analysis and tumor mutational burden of NKX2-1 expression.

(A) Violin plot of the immune score, stromal score, ESTIMATE score in NKX2-1 high-level and low-level groups. (B) Box plot showing the fractions of the 22 immune cells in NKX2-1 high-level and low-level groups. (C) Correlation between NKX2-1 expression and the 22 immune cells. Dot size indicates the correlation coefficient, with negative correlation on the left and positive correlation on the right. (D) macrophages M1 and (E) macrophages M2, (F) monocytes, (G) neutrophils, (H) mast cells resting, (I) T cells CD4 memory resting. (J) The correlation between NKX2-1 expression and tumor mutational burden. (K) Heatmap of the correlation between NKX2-1 and immune checkpoints; Pearson coefficient was used to test significance. The darker the red, the stronger the positive correlation; and the darker the blue, the stronger the negative correlation. Pearson correlation between NKX2-1 expression. *p < 0.05, **p < 0.01, ***p < 0.001.

Analysis of differences in immune therapy and pharmacotherapy responsiveness in NKX2-1 high-level and low-level groups

In NKX2-1 high-level and low-level groups, we examined the therapeutic sensitivity to chemotherapy drugs and molecular targeting drugs using the pRRophetic package and then screened out data for common clinical pharmacotherapies of cancer. The results indicated that the IC50s of various chemotherapy drugs, including 5-fluorouracil, cisplatin, docetaxel, doxorubicin, etoposide, gemcitabine, paclitaxel, and vinorelbine, and drugs for molecular targeted therapy, including axitinib, BI-2536, and gefitinib sorafenib, were lower in the NKX2-1 low-level group, indicating greater sensitivity to the above drugs (Figs. 7A–7L). We then obtained the immunotherapy score data from tumor-targeted immune cell agonist (TICA) and compared the differences in immunotherapy score between NKX2-1 high-level and low-level groups; interestingly, there was no significant difference (Figs. 7M–7P).

Figure 7 Analysis of differences in immune therapy and pharmacotherapy responsiveness in NKX2-1 high-level and low-level groups.

IC50 was calculated for (A) 5-fluorouracil, (B) axitinib, (C) BI-2536, (D) cisplatin, (E) docetaxel, (F) doxorubicin, (G) etoposide, (H) gefitinib, (I) gemcitabine, (J) paclitaxel, (K) sorafenib, (L) vinorelbine, (M) The responsiveness in combination therapy of anti-CTLA4 and anti-PD-1. (N) The responsiveness in anti-PD-1 therapy. (O) The responsiveness in anti-CTLA4 therapy. (P) The responsiveness in other immune checkpoint inhibitor therapy. ***p < 0.001.

Verification of NKX2-1 in lung cancer

Samples from the GEO database (GSE67061, GSE84784, GSE101420) were comprised of 78 LUSC samples and 69 normal lung samples. NKX2-1 expression was significantly lower in LUSC compared to normal tissue (Fig. 8A). Co-expression analysis of NKX2-1 genes and differential analysis were performed on LUSC mRNA data from TCGA and GEO databases using R software, followed by the identification of shared genes from both analyses, resulting in 1,014 genes (Fig. 8B). Co-expression analysis of the GEO database revealed positive correlations of NKX2-1 expression with SLC22A31, NAPSA, SFTA2, C16orf89, and SFTPD, while negative correlations were observed with TRIM29, GJB5, KRT5, and IRF6 (Table 4). Functional annotation through GO and KEGG pathway enrichment analysis for the 1,014 genes indicated enrichment in cell cycle processes (Figs. 8C–8F). IHC results showed NKX2-1 protein expression in tumor tissues of both LUAD and LUSC, with LUAD displaying negative, moderate, and strong expression (Figs. 9A–9C), and LUSC showing negative, weak, and moderate expression (Figs. 9D–9F). Furthermore, qRT-PCR revealed significantly lower NKX2-1 expression in LUSC compared to LUAD (Fig. 9G), with no significant difference in SFTPD and NAPSA expression levels compared to NKX2-1. However, IRF6 and TRIM29 displayed significantly higher expression levels compared to NKX2-1 (Fig. 9H).

Figure 8 Verification analysis of NKX2-1 gene in LUSC.

(A) Comparison of NKX2-1 expression level between LUSC and normal tissue. (B) Venn diagrams showing the intersection of co-expression genes of NKX2-1 and DEGs in LUSC based on GEO and TCGA databases. (C, D) GO analysis of shared genes in co-expression and DEGs. (E–F) KEGG pathway analysis of shared genes in co-expression and DEGs. ***p < 0.001.

Table 4 The co-expression genes of NKX2-1 in GEO and TCGA.

	GEO	TCGA	
Gene	Pearson’s correlation	p-value	Pearson’s correlation	p-value	
NAPSA	0.876646645	<0.0001	0.845383966	<0.0001	
SFTPD	0.806601106	<0.0001	0.813702366	<0.0001	
SLC22A31	0.833582363	<0.0001	0.873599353	<0.0001	
SFTA2	0.867622911	<0.0001	0.852154614	<0.0001	
C16orf89	0.882838078	<0.0001	0.815555528	<0.0001	
GJB5	−0.451335552	<0.0001	−0.378450636	<0.0001	
KRT5	−0.320750235	<0.0001	−0.370243082	<0.0001	
TRIM29	−0.582457298	<0.0001	−0.397324635	<0.0001	
IRF6	−0.404342555	<0.0001	−0.370549304	<0.0001	

Figure 9 Expression of NKX2-1 protein and RNA in lung cancer.

IHC results displaying NKX2-1 protein levels in LUAD (A–C) and LUSC (D–F) Based on data from the human protein atlas. Expression levels are categorized as negative (A, D), moderate (B, F), strong (C), and weak (E). Relative expression level of NKX2-1 in LUAD and LUSC (G), along with the expression levels of its co-expression genes in LUSC (H).

Discussion

NKX2-1, also known as TTF-1, is a lineage-specific transcription factor involved in the occurrence of lung cancer and regulate the adeno-to-squamous transdifferentiation to determined tumor phenotype (Mollaoglu et al., 2018; Boggaram, 2009). Many studies have shown that NKX2-1 has high sensitivity and specificity for diagnosing primary lung cancer and is expressed in most cases of LUAD (Yang et al., 2012). Additionally, NKX2-1 expression has been reported as a positive prognostic indicator for LUAD (Zhang et al., 2015). Due to the low or delection of NKX2-1 expression, the important role of NKX2-1 is neglected in LUSC (Yang et al., 2012). With the developments in cancer research bioinformatics, we can use powerful tools to analyze and explore the underlying molecular mechanisms in cancer biology and development, providing further reference value for clinical research (Banwait & Bastola, 2015). Currently, there is little research into the association between NKX2-1 expression and LUSC, so we sought to explore this relationship further using the TCGA database.

In present study, we examined and compared NKX2-1 expression between the normal and tumor tissues of several pan-cancers using TIMER data. These results indicate that NKX2-1 is mainly expressed in LUAD, LUSC, and THCA, consistent with current reports in the literature (Guan et al., 2021). Interestingly, there was no significantly difference in NKX2-1 expression between normal and tumor tissues in LUAD. We found that, in the diagnosis and treatment of LUAD, NKX2-1 expression is often detected using immunohistochemical (IHC) methods, with fewer studies examining gene expression, which is a new finding (Doherty et al., 2019; Nakra et al., 2021). Not surprisingly, the TIMER results are consistent with our results from R software, with NKX2-1 expression found to be significantly lower in tumor than in normal tissue. This is consistent with the findings in several studies that NKX2-1 expression is lower in LUSC (Yang et al., 2012; Guan et al., 2021).

Furthermore, we analyzed the relationship between NKX2-1 expression and clinicopathological features, considering any differences between NKX2-1 high-level and low-level groups, which revealed its prognostic value to a certain extent. The results of OS and PFS in K-M plotter indicated high expression of NKX2-1 is clearly linked with poor prognosis in LUSC. Meanwhile, the univariate and multivariate Cox regression analysis revealed NKX2-1 to be an independent prognostic factor in LUSC, which is consistent with the findings of Puglisi et al. (1999) The ROC curve, a graphical plot illustrating the diagnostic ability of a binary classifier system as its discrimination threshold is varied, juxtaposes sensitivity against 1-specificity. Its quantification, the AUC, is a widely recognized measure in clinical epidemiology to evaluate biomarkers’ diagnostic capabilities (Hajian-Tilaki, 2013). However, our analysis revealed that NKX2-1 expression’s prognostic utility for LUSC over 1-, 3-, and 5-year intervals fell below expectations. Historically acknowledged for its scarcity in LUSC, recent advancements in gene detection technologies, such as those employed by TCGA, have identified NKX2-1 amplification in LUSC, albeit at low levels in 2012 (Phelps, Lai & Mu, 2018; The Cancer Genome Atlas Research Network, 2012). This underexpression likely contributes to the observed diminished sensitivity in ROC curve analysis. Given the inherent limitations of ROC curves for comprehensive analysis, we employed Nomograms to further delineate NKX2-1’s prognostic significance in LUSC. Nomograms offer a personalized risk assessment, integrating clinical or disease-specific characteristics, and have been instrumental in prognostication across various cancers for years (Balachandran et al., 2015; Chen, Hu & Chen, 2020). Our study’ nomogram, incorporating both clinical features and NKX2-1 expression levels, indicates that higher NKX2-1 expression correlates with reduced OS, consistently across predicted 1-, 3-, and 5-year outcomes.

NKX2-1 is recognized for its high specificity to LUAD and serves as a crucial biomarker for its diagnosis (Yatabe et al., 2019). Nakra et al. (2021) underscores the strong link between NKX2-1 expression and EGFR mutation status, highlighting its association with favorable outcomes. Independent of EGFR mutation presence, NKX2-1 IHC positivity is correlated with improved PFS and OS (Nakra et al., 2021; Svaton et al., 2020; Ma et al., 2015). Within cancer biology, NKX2-1 plays a dual role, functioning as both an oncogenic driver and a tumor suppressor (Schilsky et al., 2017). The beneficial prognostic implications of NKX2-1 positivity in LUAD may be attributed to its anti-tumoral activities, suggesting a potential mechanism underlying its prognostic advantage (Phelps et al., 2019). Despite NKX2-1’s strong association with LUAD as opposed to LUSC, it is expressed in approximately three-quarters of LUSC cell lines, albeit not predominantly (Boggaram, 2009). A retrospective analysis by Svaton et al. (2020) highlighted the presence of NKX2-1-positive cases in LUSC, revealing a longer PFS and OS in NKX2-1-negative scenarios, aligning with findings from this investigation. Conversely, recent studies have identified a subset of LUSC cases with high NKX2-1 cytoplasmic expression, identified using the ERP8190 antibody, exhibiting enhanced OS and disease-free survival (Liao et al., 2023). This evidence suggests a prognostic and predictive significance of NKX2-1 in LUSC. With ongoing advancements in genetic testing technologies, the observed expression levels of NKX2-1 in LUSC and its impact warrant further exploration.

The occurrence and development of lung cancer is a complex and dynamic process that relies on the synergy between gene mutations and tumor microenvironment (Wood et al., 2014). The immune microenvironment is involved in the development of LUSC and that NKX2-1 is associated with lung inflammation, so we explored the relationship between the NKX2-1 expression level and immunity in LUSC (Wang et al., 2021; Maeda et al., 2011). The immune infiltration algorithm was used to evaluate the level of NKX2-1 expression with regard to immune infiltration and the distribution of immune cells. The results showed that the lower expression of NKX2-1 had the less immune infiltration in LUSC. According to CIBERSORT algorithm analysis results, the expression of NKX2-1 correlated positively with M2 macrophages, mast cells resting, neutrophils, monocytes and T cells CD4 memory resting, but negatively with M1 macrophages in LUSC. Researchers have found a link between high macrophage M2 infiltration and worse prognosis in LUSC (Han & Li, 2022). Mast cells was associated with the clinical stages of LUSC and implicated in metastasis of malignancies (Zhang et al., 2020). Monocytes has been proved the assoctiation with poor survial and metastasis in LUSC (Porrello et al., 2018). There are differences in immune microenvironment in LUSC and LUAD, particularly neutrophils (Kargl et al., 2017).Compared with LUAD, LUSC had the more enrichment of neutrophils which foster squamous cell fate (Mollaoglu et al., 2018). Loss of NKX2-1 could led to tumor-associated neutrophils recruitment to shape the immune microenvironment suitable for the survival of squamous carcinoma, which in turn promotes the development of squamous carcinoma (Mollaoglu et al., 2018). Lower infiltration of neutrophils had great prognosis (Liu et al., 2017). Lower NKX2-1 expression, lower neutrophils infiltration in our study. It may be the reason that NKX2-1 low-level group has a better prognosis.

In recent years, LUSC treatment has evolved to encompass a variety of approaches. Chemotherapy has historically been the cornerstone of LUSC therapy due to the lack of identifiable driver mutations (Qiang et al., 2022). Emerging medical advancements have facilitated the approval and clinical integration of immunotherapeutic agents for treating LUSC, offering significant patient benefits (Pan et al., 2021). Despite these developments, the prognosis for LUSC patients remains substantially suboptimal. TMB, defined as the aggregate number of mutations within the tumor genome, has demonstrated a robust correlation with responses to immunotherapy (Addeo et al., 2021; Ku et al., 2021). TMB is emerging as an evaluation method for immunotherapy and plays a vital role in immune response and as an indicator of favorable survival prognosis in LUSC patients (Yan & Chen, 2021). Evidence indicates a correlation between TMB and tumor stage, with TMB median values in LUSC showing an upward trend from Stage I to Stage IV (Zhang et al., 2020). Furthermore, Devarakonda et al. (2018) observed a favorable prognosis in patients with high TMB who underwent resection for non-small cell lung cancer. This is potentially because tumors with higher mutations are more likely to present neoantigens, rendering them susceptible to immune cell targeting, such targeting can enhance immune response activation, thereby augmenting the efficacy of immunotherapy (Zhang et al., 2020). In our investigation, a negative correlation was identified between the levels of TMB and NKX2-1 expression. This suggests that lower NKX2-1 expression may be indicative of a higher TMB, potentially correlating with a more favorable prognosis for those with diminished NKX2-1 expression. Consequently, our findings propose that LUSC cases exhibiting low NKX2-1 expression might derive more significant benefit from immunotherapeutic interventions. Above all, we believed that the expression level of NKX2-1 has certain clinical guidance significance in LUSC.

To further understand the role of NKX2-1 expression in therapy, our study was conducted to analyze the pharmacotherapy response by R package. The prediction of pharmacotherapy response through R package has been demonstrated in several clinical trials (Friedman, Hastie & Tibshirani, 2010). We used pRRophetic of R package to study the pharmacotherapy response, including chemotherapeutics and targeted drugs, will help physicians to select a suitable therapy for LUSC patients. The results indicated that the NKX2-1 low-level group was significantly more sensitive to pharmacotherapy. Immune checkpoint inhibitors (ICIs) currently considered an effective anticancer therapy for lung cancer (Liu et al., 2017). In this study, NKX2-1 expression was shown to correlate positively with immune checkpoint genes. Interestingly, the responsiveness of immune checkpoint inhibitor therapy did not significantly differ between NKX2-1 high-level and low-level groups. In the management of LUSC, chemotherapy is the cornerstone of treatment (Qiang et al., 2022). Our findings corroborate this. In LUAD, positive NKX2-1 status correlates with improved chemotherapy outcomes, and LUSC displays the opposite trend conversely (Park et al., 2019). Previous works have demonstrated that chemotherapeutic agents can enhance immune response by increasing tumor immunogenicity, similarly h-TMB elevates immunogenicity, potentially improving the efficacy of immune therapies, furthermore chemotherapy can augment TMB, suggesting a synergistic effect on tumor sensitivity to immunotherapy (Qiang et al., 2022; Kersten, Salvagno & de Visser, 2015). NKX2-1’s low expression is associated with higher TMB, indicating a heightened responsiveness to pharmacotherapy, inclusive of chemotherapy. Intriguingly, h-TMB might be amenable to immunotherapy (Qiang et al., 2022). However, our study did not observe this predicted outcome. Within the LUSC context, macrophages and neutrophils predominantly influence OS, and their interaction with ICIs is notable (Lu et al., 2023). Our analysis identified macrophages and neutrophils as the immune cells linked with NKX2-1 expression, possibly elucidating the lack of association between NKX2-1 expression and immunotherapeutic efficacy. A high macrophage density suggests a ‘cool’ tumor state, characterized by low TMB and reduced immunotherapy susceptibility (Qiang et al., 2022; Lu et al., 2023). Given that low NKX2-1 expression corresponds to both lower macrophage density and higher h-TMB, it is plausible that increasing TMB, in response to chemotherapy, could render LUSC more receptive to immunotherapy.

Due to the complex oncogenic mechanisms involving NKX2-1, we explored the DEGs in both NKX2-1 high-level and low-level groups and identified co-expression genes. NKX2-1-AS1, SLC22A31, NAPSA, SFTA2, C16orf89, SFTPD, TRIM29, LINC01980, GJB5, KRT5, and IRF6 were found to be correlated with NKX2-1. NKX2-1-AS1, NAPSA, SFTA2, SFTPD, TRIM29, KRT5, and IRF6 had previously been identified to be associated with lung cancer development (Phelps, Lai & Mu, 2018; Xiao et al., 2017; Davé, Childs & Whitsett, 2004; Xu et al., 2020; Liu et al., 2021). To explore the biological mechanisms of DEGs in NKX2-1 high-level and low-level groups, we conducted GO function and KEGG pathway enrichment analyses. Based on enrichment and GO function, the DEGs were mainly found to be involved in humoral immunity. KEGG pathway analysis further indicated the DEGs were mainly enriched in cytokine-cytokine receptor interaction, cAMP signaling pathway, viral protein interaction with cytokine–cytokine receptor, and PPAR signaling pathway categories. The cAMP and PPAR signaling pathway categories are known to be closely linked with LUSC carcinogenesis and development (Ma et al., 2021). GSEA revealed that the downregulation of NKX2-1 was involved in tumor progression of LUSC, suggesting that it may be important for the therapeutic benefits of LUSC.

To validate our findings, we analyzed the mRNA data of LUSC using the GEO database. The results from GEO were in agreement with those from TCGA, showing a significant down-regulation of NKX2-1 expression in LUSC compared to normal lung tissue. The discrepancies in the number of DEGs and NKX2-1 co-expressed genes in LUSC, obtained using R software, may be attributed to potential variations in the techniques employed for data collection between the GEO and TCGA databases. Then we performed a cross-analysis of differential genes and co-expressed genes obtained from both databases. The resulting common genes were subjected to GO and KEGG enrichment analyses, which further supported the findings of GSEA enrichment analysis in our study. The enrichment analyses indicated that the involvement of NKX2-1 in LUSC primarily revolves cell cycle regulation. Previous studies have demonstrated that NKX2-1 directly regulates the cell cycle, by controlling over the expression of proliferation-related genes (Tagne et al., 2012). Several studies have demonstrated that the oncogenic mechanism of NKX2-1 involves numerous signaling pathways, such as the AKT, p38 signaling, PI3K, and WNT signaling pathways and uncovered direct transcriptional targets LMO3, EGFR, SOX2, and DUSP6 (Mollaoglu et al., 2018; Phelps, Lai & Mu, 2018; Ingram et al., 2022; Zewdu et al., 2021). Harada et al. (2017) demonstrated that NKX2-1 binds to cyclin D1 (CCND1) and plays a role in cell cycle progression, meanwhile over expression of NKX2-1 leads to increased cyclin D1 (CCND1) levels, potentially influencing metastasis incidence. It is likely to contribute to the relatively poor prognosis associated with high NKX2-1 expression in LUSC.

The Human Protein Atlas (HPA) provides valuable information on the protein levels of human gene expression profiles in both normal and tumor tissues (Navani, 2016). Through immunohistochemistry data available in this database, we observed NKX2-1 expression not only in LUAD but also in LUSC. These findings are consistent with some existing immunohistochemical studies (Ordóñez, 2012). Our analysis using TIMER data revealed a lower expression level of NKX2-1 in LUSC compared to LUAD. These findings were further supported by qRT-PCR results, indicating a consistent trend. The observed disparity in expression levels between LUAD and LUSC raises the possibility that NKX2-1 might exhibit strong expression in LUAD but only moderate expression in LUSC, warranting further investigation. Further exploration of co-expressed genes, such as NAPSA, SFTPD, TRIM29, and IRF6 in LUSC cell lines (H520) through qRT-PCR correlated with our findings, providing additional insights into the mechanisms involving NKX2-1 in LUSC development. Due to its low expression in LUSC and it is considered as a marker for identifying LUAD (Yang et al., 2012), research on NKX2-1 in LUSC has been limited, and the related mechanisms have not been fully elucidated.

Lineage plasticity contributes to the complexity of intratumoral heterogeneity, facilitates histological transitions among tumor subtypes, and may underlie the mechanisms of resistance to therapeutics observed in lung cancer (Quintanal-Villalonga et al., 2020). NKX2-1 is instrumental in regulating transcriptional programs within the pulmonary domain and is expressed not only in LUAD but also in LUSC (Mollaoglu et al., 2018). The upregulation of USP13 during early lung tumorigenesis has been reported to suppress NKX2-1 expression while enhancing SOX2 expression, thus fostering LUSC development (Kwon et al., 2023). Interestingly, LUAD has the potential to undergo histological transformation to LUSC, which may confer resistance to targeted therapies (Kwon et al., 2023). Given the critical role of NKX2-1 in LUSC, its study is of considerable importance. An analysis of NKX2-1-associated differentially expressed genes revealed an association with humoral immunity, aligning with the TME analysis. This association may explain why a significant subset of LUSC patients does not benefit from immunotherapy. Conversely, chemotherapy has been shown to elevate the TMB, and patients with a high TMB could be more responsive to immunotherapy. Therefore, a combination of chemotherapy and immunotherapy emerges as a pragmatic treatment strategy for LUSC. Studies support this approach, indicating improvements in median progression-free survival, overall survival, and response rates compared to chemotherapy alone (Qiang et al., 2022; Lu et al., 2023; Yu et al., 2022). In light of these findings, the intricate role of NKX2-1 in LUSC and its implications for pharmacotherapy necessitate additional clinical and foundational research.

In our study, we comprehensively illustrate the importance of NKX2-1 in LUSC, and validated by GEO database, HPA database and qRT-PCR. However, our study has some limitations. First, our study is based on public databases, although we conducted preliminary experiments to verify, the in-depth experimental verification is still thus lacking. Second, NKX2-1 in LUSC are rarely reported in literature, lack of literature references for us. Finally, lack of further experiments in vitro and in vivo verified the biological mechanisms of NKX2-1 in LUSC. Above all, the NKX2-1 requires further in-depth study for LUSC.

Conclusion

In conclusion, this study demonstrated that low NKX2-1 expression is closely associated with increased survival and favorable outcomes in terms of disease progression. The immune indicators of immune infiltration cells, TMB, and immune checkpoint genes were shown to be related to NKX2-1 expression level, inferring that NKX2-1 probably affects LUSC development via the TME. We also explored the responses to pharmacotherapy and immune checkpoint inhibitor therapy to offer robust new evidence for the development of potential LUSC therapies and diagnostic methods. Furthermore, validation was performed using the GEO databases, HPA databases, and qRT-PCR.As a result, we provide evidence demonstrating that NKX2-1 is a potential target for the treatment of LUSC.

Supplemental Information

Supplemental Information 1 Supplementary materials.

The manuscript has been reviewed and approved by all authors. We extend our gratitude to Xiaoying Huang’s team from the Division of Pulmonary Medicine, the First Affiliated Hospital of Wenzhou Medical University for their valuable support and assistance. We also acknowledge The Cancer Genome Atlas (TCGA) and The Gene Expression Omnibus (GEO) database for providing their platforms, and all the contributors for sharing their meaningful datasets.

Additional Information and Declarations

Competing Interests

Author Contributions

Data Availability

The authors declare that they have no competing interests.

Huiyue Lin conceived and designed the experiments, performed the experiments, analyzed the data, prepared figures and/or tables, authored or reviewed drafts of the article, and approved the final draft.

Juyong Wang conceived and designed the experiments, performed the experiments, authored or reviewed drafts of the article, and approved the final draft.

Qing Shi analyzed the data, prepared figures and/or tables, authored or reviewed drafts of the article, and approved the final draft.

Minmin Wu performed the experiments, analyzed the data, prepared figures and/or tables, authored or reviewed drafts of the article, and approved the final draft.

The following information was supplied regarding data availability:

The raw data are available in the Supplemental File and the R code is available at GitHub: https://github.com/lannyRcode/LUSC-Rcode.

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
