# Peer review of "Significance of NKX2-1 as a biomarker for clinical prognosis, immune infiltration, and drug therapy in lung squamous cell carcinoma"

_PeerJ, doi:10.7717/peerj.17338_

## Round 0.1 · original submission · Major Revisions

Dear Authors,

in the attachment, we are sending the reviewer's report, according to which it is necessary to correct the paper.

**Language Note:** The review process has identified that the English language must be improved. PeerJ can provide language editing services - please contact us at copyediting@peerj.com for pricing (be sure to provide your manuscript number and title). Alternatively, you should make your own arrangements to improve the language quality and provide details in your response letter. – PeerJ Staff

Reviewer 1 ·

Basic reporting

In this manuscript, the authors aimed to determine the prognostic value of NKX2-1 in lung squamous cell carcinoma (LUSC) using bioinformatic tools. They observed reduced NKX2-1 expression in LUSC compared to normal tissue, with variations by gender, stage, and N classification. Elevated NKX2-1 expression correlated with shorter survival, identified as an independent prognostic factor. Differential gene expression analysis revealed 51 upregulated and 49 downregulated genes in NKX2-1 high-level groups, associated with cell cycle and DNA replication. Tumor microenvironment analysis showed connections between NKX2-1 expression and immune cells. Pharmacotherapy sensitivity was higher in NKX2-1 low-level, while immunotherapy response showed no significant difference. GEO data analysis and immunohistochemistry confirmed NKX2-1 protein expression in lung adenocarcinoma (LUAD) and LUSC tissues. It is an interesting study to identify better treatment strategy for LUSC, however, this manuscript will need some revisions..

1) There are grammatical and sentence formation errors, authors should proofread the document again.
2) Figures are hard to read, authors should submit the high resolution images.
3) Could the authors provide insights into the comparison of the prognostic significance of NKX2.1 expression levels in conditions such as lung adenocarcinoma (LUAD) versus lung squamous cell carcinoma (LUSC)?
4) Can the authors offer insights into the therapeutic approach for cases with higher levels of NKX2.1 compared to those with lower levels, especially considering that lower level groups are more sensitive to pharmacotherapy?

Experimental design

No comment

Validity of the findings

No comment

Additional comments

No comment

Reviewer 2 ·

Basic reporting

No comment

Experimental design

The methods described specifically the R libraries used have been correctly mentioned in the text. However, to help the readers to reproduce the results, details about the commands used can be added as a supplementary file.
The results obtained for the nomogram and respective AUC of the ROC curve is on the lower end (0.54-0.57) which would lead to uncertainty in the prediction. Can the authors explain these AUC values and if they can improve on these values by either eliminating or adding additional features.

Validity of the findings

No comment

Reviewer 3 ·

Basic reporting

Nothing.

Experimental design

Nothing.

Validity of the findings

Nothing.

Additional comments

This study investigated the function of NKX2-1 in LUSC, which has not received much attention so far, using various tools and datasets. The findings in this study are considered very important in that NKX2-1 is a potential biomarker and therapeutic target for LUSC. However, there are several concerns that need to be revised for publication.

Major comments:

1. In the introduction, it is mentioned that NKX2-1 expression is extremely low in LUSC, but in Fig. 1, NKX2-1 expression does not appear to be that low. How do you interpret this discrepancy?
2. In this study, the authors found a negative relationship between TMB and NKX2-1 expression. Please add more discussion on the purpose of conducting this analysis and your interpretation of these findings.
3. Authors conducted various analyses on NKX2-1 in this study. Could you integrate those results and add a discussion on the function and mechanism of NKX2-1 in LUSC.

Minor comments:

1. The authors mentioned the use of microarray data in Abstract section and RNA-Seq data in Materials and Methods section. Which is correct?
2. In lines 130-136, how were the RNA-Seq raw data mapped and counted to calculate FPKM? Could you clarify the tools and procedures used for this analysis.
3. What is the definition of "tumor-specific mutated genes" mentioned in line 171? How many genes are included?
4. Were multiple comparisons performed in Fig. 2C, D, and F? If not, it is appropriate to do so.
5. Please cite Fig. 3 and Fig. 6 in alphabetical order in the Results section.
6. How were the 729 DEGs identified in line 264? Are these different from the 51 and 49 genes mentioned in the previous section?
7. The text in Fig. 4 is not visible. Please replace it with a higher-resolution image.
8. What does the y-axis in Fig. 8A represent? Please clarify this on the graph.
9. How were the nine genes of interest (Table 4) identified in lines 323-324?
10. There is a typo in line 348; "LUSA" should be "LUSC".

---

## Round 0.2 · accepted · Accept

The revision addresses the reviewers' suggestions.